# Novel Epitopes Mapping of African Swine Fever Virus CP312R Protein Using Monoclonal Antibodies

**DOI:** 10.3390/v15020557

**Published:** 2023-02-17

**Authors:** Yibrah Tekle Hagoss, Dongdong Shen, Zhenjiang Zhang, Fang Li, Zhigao Bu, Dongming Zhao

**Affiliations:** 1State Key Laboratory of Veterinary Biotechnology, Harbin Veterinary Research Institute, Chinese Academy of Agricultural Sciences, Harbin 150069, China; 2Department of Animal and Range Sciences, College of Agriculture and Natural Resources, Raya University, Maichew P.O. Box 92, Ethiopia

**Keywords:** ASFV CP312R, epitope mapping, immunogenic, monoclonal antibody, subcellular location

## Abstract

African Swine Fever (ASF) is a highly contagious and lethal pig disease and poses a huge threat to the pig industry worldwide. ASF virus (ASFV) encodes more than 150 different proteins, but the biological properties of most viral proteins are still unknown. ASFV CP312R protein has been proven to be one of the most immunogenic proteins during ASFV infection in pigs; however, its specific epitopes have yet to be identified. In this study, we verified the immunogenicity of CP312R protein in the sera from attenuated ASFV-inoculated pigs. We generated seven anti-ASFV CP312R mouse monoclonal antibodies (mAbs) from mice immunized with recombinant CP312R protein (rCP312R). All seven mAbs are the IgG2b-Kappa isotype and specifically interacted with the CP312R protein expressed in various cells that were infected by ASFVs or transfected with plasmid CP312R. The epitope mapping was performed by using these characterized mAbs and the peptide scanning (Pepscan) method followed by Western blot. As a result, two antigenic determinant regions were identified: two of the seven mAbs recognized the ^122^KNEQGEEIYP^131^ amino acids, and the remaining five mAbs recognized the ^78^DEEVIRMNAE^87^ amino acids of the CP312R protein. These antigenic determinants of CP312R are conserved in different ASFV strains of seven genotypes. By using the characterized mAb, confocal microscopy observation revealed that the CP312R was mainly localized in the cytoplasm and, to some extent, in nuclei and on the nuclear membrane of infected host cells. In summary, our results benefit our understanding on the antigenic regions of ASFV CP312R and help to develop better serological diagnosis of ASF and vaccine research.

## 1. Introduction

African Swine Fever (ASF) is a highly contagious and lethal hemorrhagic viral disease of wild and domestic swine of all ages and breeds that has a wide range of clinical symptoms [1,2]. This economically important pig disease was documented for the first time in 1921 in Kenya, East Africa [3]; since then, it has disseminated to more than 50 countries on all 5 continents (Asia, Africa, America, Oceania, and Europe) [2,4,5,6,7]. Its’ main modes of transmission include direct contact, indirect contact with fomites, consuming infected meat or meat products, and soft ticks [2].

African swine fever virus (ASFV) is the etiological agent of ASF, which is a large dsDNA arbovirus [8,9,10]. It is the only member of the *Asfarviridae* family in the *Asfivirus* genus [9] that encodes more than 150 different proteins which are involved in the structure, immune evasion, and DNA replication of the virus [11,12]. Most of the ASFV genes are nonessential genes and do not participate in ASFV replication [11]; however, they do contribute to the control of virus–host association, pathogenesis, and immune response [1].

The presence of inadequate information on host–virus interactions, limited data on mechanisms that defend host immunity, the complexity of the virus genome, and the presence of many uncharacterized genes have hindered the development of ASFV vaccines or treatment options [1,13,14,15,16]. Early pathogen detection is important when a disease lacks a vaccine as part of its prevention strategy. Thus, researchers have attempted to develop various serological diagnosis kits by using monoclonal antibodies (mAbs) against the ASFV immunogenic proteins such as [17,18] p. 72, [19] p. 62, [20,21,22,23] p. 54, [22,24] p. 30 , [25] p. 22 and others.

The mAbs’ development has shown a significant increase in the areas of modern medicine, from their application for diagnostic purposes to the design of vaccines and/or treatments for diseases. To date, about 100 mAbs have been approved by the FDA, and the annual global market report for 2021 was $150 billion [26,27]. In addition, mAbs are useful for probing protein structure and function and for identifying and purifying proteins (as analytical markers).

CP312R is one of those understudied ASFV genes [15]. The existing research only reports that it is immunogenic in a pooled viral vector expression [28,29,30], and based on its crystal structural analysis, it has single-stranded DNA binding activity [31]. This study initially verified the target antigen’s capability of inducing the immune system. Then, we generated and characterized a panel of seven anti-ASFV CP312R mouse mAbs, and by using these characterized mAbs, this study identified the conserved ASFV CP312R epitopes. The epitopes are mainly located at ^78^DEEVIRMNAE^87^ and ^122^KNEQGEEIYP^131^ AA residues of the target CP312R protein. We also investigated the ASFV CP312R subcellular location during the virus infection in its host, which was mainly located in the cytoplasm and, to a lesser extent, in the nuclei and on the nuclear membrane of the infected cells.

## 2. Materials and Methods

### 2.1. Ethics Statement

The guide for the care and use of laboratory animals published by the Ministry of Science and Technology of the People’s Republic of China was strictly followed during the animal experiment on mice, which was approved by the Animal Care and Use Committee of the Harbin Veterinary Research Institute (HVRI) of the Chinese Academy of Agricultural Sciences (CAAS).

### 2.2. Cells Maintenance

The SP2/0 myeloma cells were maintained in Roswell Park Memorial Institute (RPMI)-1640 medium that was supplemented with 10% heat inactivated fetal bovine serum (FBS), 1% L-glutamine, 1% penicillin and streptomycin, and 0.2% fungin, anti-fungal (InvivoGen, San Diego, CA, USA). Similarly, the hybridoma cells (SP2/0 myeloma fused with mouse B-cells) were cultured in an RPMI-1640 medium that was supplemented with some changes, such as 20% FBS and HAT 50X. This medium contains HAT (Hypoxanthine-Aminopterin-Thymidine), which was discarded after seven days and replaced with HT (Hypoxanthine-Thymidine) 50X supplement RPMI-1640 medium. Then, a week later, the hybridoma cells were cultured in RPMI-1640 medium with 10% fetal bovine serum, as previously described [32].

Also, PAMs were maintained in the supplemented RPMI-1640 medium which included 10% porcine serum. The WSL, HEK-293T, and Vero cells were cultured in Dulbecco’s modified Eagle’s medium (DMEM) that was supplemented with heat-inactivated fetal bovine serum 10% (*v*/*v*), antibiotics 1% (penicillin and streptomycin sulfate), L-glutamic amino acids 1% (Invitrogen, Portland, OR, USA), and anti-fungal 0.1%. All the cell lines were incubated at 37 °C in the presence of 5% CO_2_.

### 2.3. Antibodies and Reagents

The study used different antibodies that were purchased from various companies, such as the mouse primary antibodies against His-tag (66005-1-Ig, 1:5000) from Proteintech (China); HRP-conjugated goat anti-mouse IgG antibody (12-349, 1:10,000) from Sigma-Aldrich (Saint Louis, MO, USA); HAT media supplement 50X and complete and incomplete adjuvants from Sigma-Aldrich; HT media supplement 50X from Beyotime Co., Huaian, China; fluorescein isothiocyanate (FITC, 1:100) from ThermoFisher; and Hoechst 33342 (1:10,000) staining dye solution from Abcam (Waltham, MA, USA) were among the reagents.

### 2.4. Plasmids Construction

A CP312R sequence (924 bp) was generated from the Pig/HLJ/2018 isolate of ASFV genotype II (GenBank: MK333180) [33]. For purification purposes, we have allowed incorporating the 6x-His tag with our target CP312R protein, and the plasmid PET-30a (+) was digested with Xho I and EcoR I endonuclease restriction enzymes. For transfection purposes, the full length sequence of the CP312R was also cloned into the pCAGGS without any tag. Similarly, the plasmid was digested by the Xho I and EcoR I enzymes. Furthermore, for epitope mapping, all the fragment primer sets (Table 1) were designed according to the ClonExpress II One Step Cloning Kit (C112, Vayzme, Biotech) by incorporating the vector’s upstream and downstream homologous sequences (15–20 bp) into the forward and reverse primers, respectively, without using the endonuclease restriction enzymes. The C-Myc-tag fused truncates were cloned into PCAGGS vector, which was digested with the restriction enzymes Xho I and EcoR I. The ligated genes were transformed into *E. coli* DH5 Alpha-competent cells on kanamycin (pET-30a (+))- or ampicillin (pCAGGS)-supplemented agar plates overnight in a 37 °C incubator. The recombinant DNAs were confirmed by sequencing and alignment.

### 2.5. Expression and Purification of Recombinant CP312R Protein

A 6x His-tagged fusion CP312R protein was expressed in *E. coli* BL21 (DE3) competent cells and was found in adequate amounts in the supernatant of the lysed bacteria after 6 h of induction by 1M IPTG (isopropyl-β-D-1-thiogalactoside). This pilot survey demonstration was scaled up to large volume bacterial production using the developed protocol. Subsequently, the collected bacteria (8000 rpm for 10 min at 4 °C) were suspended in cold 1 X PBS to prepare for sonication. The harvested supernatant was filtered and purified using the affinity-based AKTA Avant liquid chromatography purification system. Then, Western blotting analysis was performed to determine the quality of the purified recombinant CP312R protein by using the anti-His tag antibody.

### 2.6. Immunization, Cells Fusion and Production of Monoclonal Antibodies

To produce the mAbs against ASFV CP312R, this study customized the previously described protocols [32,34]. The 6–8-week-old Balb/c mice were immunized subcutaneously with 100 µg/mouse of the purified ASFV CP312R protein mixed with an equal volume of complete Freund’s adjuvant, and then, every 14 days, two booster doses of the protein mixed with incomplete Freund’s adjuvant were injected. The mouse antibody titration was measured by an indirect ELISA and Western blotting assay. The positive mouse was selected for a further 100 µg/mouse of purified ASFV CP312R protein (without adjuvant) intraperitoneal (i.p.) injection. After 2–3 days, the mouse’s splenocytes (B cells) were collected to fuse with SP2/0 (myeloma cells) at a ratio of 10:1 in the presence of polyethene glycol (PEG) 1450 [35]. The high-titer clones were chosen for further subcloning until a single clone was generated; meanwhile, they were subjected to subcloning by limited dilution [36,37] four times. As a result, a panel of seven anti-CP312R mouse mAbs was produced.

### 2.7. Monoclonal Antibody Isotyping

The immunoglobulin subclass or isotype of the anti-ASFV CP312R mAbs (single-clone positive hybridoma cells) was characterized by using the Pierce Rapid ELISA Mouse mAb Isotyping Kit (ThermoFisher Scientific, Waltham, MA, USA), as per the manufacturer’s instructions.

### 2.8. Indirect-Enzyme-Linked Immunosorbent Assay (iELISA)

It was conducted to screen the positive hybridoma clones and their specificity and immunogenicity for CP312R, as previously stated [36]. The 96-well microtiter plate (Nunc) was coated with purified recombinant CP312R protein or other target antigens (0.3 µg/well), which was then incubated overnight at 4 °C. After being washed three times with PBST (0.05% Tween 20 in PBS), the plate was blocked with 5% nonfat milk (200 µL/well) in PBS for 1 h at 37 °C. Subsequently, the plate was incubated with the supernatant of the hybridoma or sera (1:200 in PBS) (50 µL/well) for 1 h at 37 °C. Next, three washes were followed by an HRP-conjugated anti-mouse (10,000) or anti-pig (1:20,000) secondary antibody (Sigma-Aldrich) for 1 h at 37 °C. The washed plate was incubated with 50 µL/well of the chromogenic substrate solution 3,3′,5,5′-tetramethylbenzidine (TMB) for 15 min in the dark. Finally, the reaction was stopped by 2 M H_2_SO_4_ (50 µL/well) and then measured using a Bio-Rad microplate reader at OD450 nm absorbance. Each sample was analyzed in triplicate and assumed to be positive when its average measured OD450 nm value was higher than the negative control’s average OD450 nm value.

### 2.9. Western Blotting Assay

It was used to determine the immunogenicity of CP312R, investigate the specificity of the anti-ASFV CP312R mAbs produced against the purified ASFV CP312R recombinant protein, and perform epitope mapping as previously described [37]. The lysates or purified recombinant ASFV CP312R protein was resolved by using 10% or 12% Sodium Dodecyl Sulfate–Polyacrylamide Gel Electrophoresis (SDS-PAGE) (80 V for 20–30 min and then 140 V for 60 min) to transfer onto polyvinylidene difluoride (PVDF) membranes (Millipore, Billerica, MA, USA) for 90 min at 10 V. The PVDF was blocked by 5% nonfat milk in PBST (0.05% Tween 20 in PBS) for 1 h at room temperature (RT). Then, the membrane was incubated for 2 h at RT with primary antibodies, pig sera (1:100), anti-ASFV-CP312R mAbs (1:100), and an anti-His-tag antibody. Subsequently, after being washed three times, the membrane was incubated with the HRP-conjugated anti-mouse (1:10,000) or pig (1:20,000) secondary antibodies. The protein band was observed using the SuperSignal West Femto (ThermoFisher, Waltham, MA, USA) enhanced chemiluminescence (ECL) reagent and the Azure Biosystems reader machine.

### 2.10. Immunofluorescence Assay (IFA)

Both HEK-293T cells and PAMs were used for IFA, transfected with full-length or truncated CP312R plasmids, or infected with Pig/HLJ/2018 [38], respectively. This assay followed the protocol mentioned previously [37], with some modifications. The cells were cultured in 24-well plates for 24 h before being fixed (4% paraformaldehyde) and permeabilized (0.25% Triton X-100) for 25 min at RT. After being blocked by 0.25% BSA for 1 h at RT, the cells were incubated with each anti-ASFV CP312R mouse mAb for 2 h at RT. The cells were incubated with an anti-mouse FITC dye-labeled secondary antibody (ThermoFisher Scientific; 1:100) for 50 min, and then the washed cells were mounted to visualize the fluorescent signals.

### 2.11. Epitope Mapping of CP312R

The antigenic determinant regions of CP312R were identified with the characterized anti-ASFV CP312R mAbs and the peptide scanning (PepScan) method, where multiple series of overlapping peptides of CP312R were designed and synthesized, followed by the Western blotting assay [39,40].

### 2.12. Lentivirus Production

Lentivirus packaging has been performed according to the TransIT-lentivirus transfection kit (Mirus Bio, Madison, WI, USA) by using HEK-293T cells as per the company’s instructions. The combined lentivirus packages (10 µL, 1 µg/µL) and plasmid CP312R (1 µg) were mixed with 200 µL of Opti-MEM I Reduced-Serum Medium, and then 6 µL of TransIT-Lenti Reagent was added into the diluted DNA mixture and kept at RT for 10 min. Then, after 48 h, the supernatant (lentivirus) was harvested and centrifuged (300× *g* for 5 min). The harvested and filtered lentivirus-CP312R infected the WSL cells, and the effective lentivirus packaging was confirmed by the Western blotting assay.

### 2.13. Subcellular Localization of CP312R

The PAMs and WSL cells, respectively, were infected with ASFV HLJ/HRB1/2020 and lentivirus-CP312R; then, confocal microscopy was used for subcellular location analysis. The experimental procedures were monitored as previously described by Zhong et al. [41], with some modifications. The characterized anti-CP312R mouse mAb-C4 was used as a primary antibody. Finally, the cells were incubated with Hoechst 33342 nuclear stain for 15–20 min at RT to visualize where the CP312R target protein is localized.

### 2.14. Cell Transfection

The confluent HEK-293T and Vero cells in a six-well plate were transfected by using the TransIT-293 reagent (Mirus Bio, Madison, WI, USA) as per the manufacturer’s protocol. For a single well of the six-well plate, 7.5 µL of TransIT-293 Reagent was first mixed with 2.5 µL of plasmid DNA (1 µg/µL), and then kept at room temperature for 5 min. After that, it was thoroughly mixed with 250 µL of Opti-MEMI Reduced-Serum Medium. Finally, after 25 min at room temperature, 260 µL per single well from the complex was added dropwise. Subsequently, cells were incubated for 48 h at 37 °C in the presence of 5% CO_2_.

### 2.15. Statistical Analyses

The experimental results were derived from representative experiments that were repeated at least three times. Microsoft Excel 2016 and GraphPad Prism 8.4.3 (GraphPad Software, San Diego, CA, USA) were used to summarize and generate the graphs.

## 3. Results

### 3.1. Expression and Purification of the Recombinant CP312R Protein

The recombinant His-tagged CP312R protein was expressed in *E. coli* BL21 (DE3) at 37 °C with 1 mM IPTG induction for 6 h until the bacteria’s optical density (OD) reached 0.6, at which point it was expressed in both the supernatant (soluble) and pellet sections of the sonicated bacteria (Figure 1A). It was purified by using a 6x-His Trap HP column (Cytiva, Marlborough, MA, USA) in an AKAT affinity-based liquid chromatography purifying system, in which the product was subjected to SDS-PAGE for Coomassie Brilliant Blue staining (CBB). The sole peak of the purification process was in the range of 85–105 mL of elution volume (Figure 1B). The CP312R immunogenicity was determined by the detection of the antibody response against the CP312R in the sera from pigs infected with the ASFV Pig/HLJ/HRB1/2020 strain by using the iELISA and Western blotting assays, in which the protein was found bound to the sera’s antibody in both assays (Figure 1C).

### 3.2. Production and Characterization of Anti-ASFV-CP312R Monoclonal Antibodies

To generate the specific mAbs against the CP312R, the traditional production method was used, in which we immunized three female Balb/c mice with the purified recombinant CP312R protein. The iELISA and Western blotting assays were used to select the positive mouse, and the results indicated that all the mice responded to the antigen protein (Figure 2A). Then, a panel of seven anti-ASFV CP312R mouse mAbs was produced after being subcloned four times and screened. They were named mAb-C4, mAb-C5, mAb-B2, mAb-B11, mAb-D7, mAb-E4, and mAb-G11. The iELISA test showed that the mAbs scored a higher antibody binding titer (OD450 nm) than the negative and positive controls (Figure 2B). Their isotyping indicated that all the mAbs were identified as subclass IgG2b/κ-type (Figure 2C).

The mAbs were characterized by using the indirect immunofluorescence assay (IFA) in the cells of the HEK-293T and PAMs, which had been transfected with the plasmids of CP312R (1µg/well, 12-well plates) and infected with the ASFV Pig/HLJ/2018 isolate (MOI = 1) of the ASFV genotype II, respectively. As a result, the assay showed that all mAbs interacted with the expressed antigen in the HEK-293T cells (Figure 3A) or PAMs (Figure 3B). The mock control in cell lines exhibited no fluorescence. In addition, to determine the specific reactivity of mAbs with the recombinant CP312R protein that was expressed in various cells, we carried out the Western blotting analyses using the lysates collected from the Vero and HEK-293T cell lines that were transfected with plasmid CP312R, from PAMs that were infected with pig/HLJ/HBR1/2020 strain of ASFV, and from WSL cells that were infected with the lentivirus CP312R. The mAbs showed reactivity with the recombinant protein in those cells (Figure 3C). Furthermore, to confirm whether the anti-ASFV CP312R mouse mAbs had cross-reactivity with other proteins, we performed the iELISA and Western blotting analyses using other purified antigens that were available in our laboratory, such as ASFV A104R, porcine-HEV, PLCP, swine MBP, and CP312R. The analyses showed that, except for CP312R, none of the mAbs recognized them in Western blot (Figure 3D) or iELISA (Figure 3E).

### 3.3. Epitopes Mapping of Recognized by the Monoclonal Antibodies

To explore the antigenic determinants of the CP312R using the characterized mAbs, the CP312R protein sequence was truncated to a series of various overlapped regions (Figure 4A) and confirmed by Western blotting assay. Accordingly, CP312R was segmented into ten fragments with amino acid residues spanning 1–152, 1–141, 1–130, 1–119, 1–108, 1–100, 1–86, 1–75, 1–64 and 153–307, and each fragment was tested with each mAb. The analysis results showed that the mAb-C4, mAb-C5, mAb-B11, mAb-E4, and mAb-G11 only interacted with the segments spanning AA 1–152, 1–141, 1–130, 1–119, 1–108 and 1–100 (Figure 4B, upper panel), while the mAb-D7 and mAb-B2 were bound to the segments spanning only aa 1–152 and 141 (Figure 4B, lower panel). Furthermore, five truncates from each of the candidate CP312R regions were designed and constructed to determine the right position of the epitope through trimming AAs from both sides (Figure 4A). The Western blotting assays indicated that the two mAbs (mAb-B2/D7) reacted with the expressed fragments of 1–131 and 122–307, but not with 132–307, 123–307, or 124–307 (Figure 4C, lower panel). Similarly, the five remaining mAbs bound specifically with the expressed truncations of the CP312R protein’s 1–87 and 78–307 AA residues, but not with the 79–307, 80–307 and 88–307 residues (Figure 4C, upper panel). Finally, based on the mapping analysis results, two linear antigenic determinants of ASFV CP312R were identified, which are accurately positioned at ^122^KNEQGEEIYP^131^ (mAb-D7 and mAb-B2) and ^78^DEEVIRMNAE^87^ (mAb-C4, C5, B11, G11, and E4) AA residues of the CP312R protein.

The IFA analysis of these two antigenic determinants using harbor plasmids (1–100 AA for mAb-D7 and B2, and 1–141 AA for the remaining five mAbs) in HEK-293T cells allowed the epitopes to be recognized by the five mAbs (Figure 4D, upper panel) as well as by mAb-D7 and mAb-B2 (Figure 4D, lower panel).

Finally, the sequence alignment analysis demonstrated that the identified antigenic determinants of the CP312R protein are highly conserved across seven ASFV genotypes except for genotype VIII (Figure 5).

### 3.4. Subcellular Location of CP312R

Using one of the characterized anti-ASFV CP312R mAbs (mAb-C4), this study investigated the CP312R protein location in PAMs infected with the pig/HLJ/HRB1/2020 strain of ASFV, and in WSL cells infected with lentivirus-CP312R. The fixed (4% paraformaldehyde) cells were permeabilized (0.25% Triton X-100) to make them accessible to the anti-CP312R mAb. The confocal microscopy analysis indicated that the ASFV CP312R protein is located mainly in the cytoplasm and, to some extent, in the nuclei and on the nuclear membrane of the infected WSL and PAMs (Figure 6).

## 4. Discussion

ASFV is a potential intercontinental pathogen that has been found in more than 50 countries worldwide [2,5,6,7] and has continued to cause colossal economic losses [5,42]. Currently, it is an unpreventable virus that lacks vaccines and drug options [43,44] due to its genetic complexity, insufficient information about molecular mechanisms of pathogenesis, many genes that are still poorly studied, and the variability of the virus’s isolates [1,15,16,42,45,46]. Even though it is still poorly defined [15], very few previous reports have identified the CP312R protein as a vital potential target protein for the development of a vaccine or inhibitor drug against the ASFV infection. Its ability to mount antigen-specific responses to virally vectored CP312R [28,47] and its SSB core site [31] may contribute to the prevention of ASFV DNA replication during the virus’s infection in host cells.

Using sera from pigs infected with the highly virulent Pig/HLJ/HRB1/2020 strain of ASFV [38] in iELISA and Western blotting analyses, our study has proven that the CP312R protein is an immunogenic ASFV protein, which is in line with the previous reports of CP312R [28,47]. According to earlier studies, the immunogenic ASFV proteins, such as the p. 30, p. 54, p. 62 and p. 72 proteins, were used either alone or in combination in serological detection assays and in the production of vaccines [15,28,47,48,49,50]; CP312R can also be used.

This study used purified CP312R protein for the mAb preparations by immunizing Balb/c mice for the purposes of epitope mapping and subcellular location analyses. As a result, a panel of seven anti-ASFV CP312R mouse mAbs was generated. Subsequently, the mAbs were characterized by using various cells infected by the ASFV and lentivirus-CP312R as well as by plasmid CP312R transfected in the assays of Western blot and IFA, in which all the mAbs were specifically reactive with the CP312R protein. Furthermore, except for the control (CP312R) protein, none of the different purified proteins used in the iELISA and Western blotting analyses showed interaction with the mAbs. Since the CP312R protein is still undescribed [15,50], these characterized anti-CP312R mAbs may be used as research reagents in studies of expression, functions, and pathogenic molecular Mechanism.

Because of their high specificity, mAbs are effective instruments for determining the precise location of antigen epitopes [49]. Epitopes (antigenic determinants) are chemical groups on an antigen’s immunoactive surface that are selectively bound by antibodies. Epitopes are vital elements in evaluating the viral proteins’ antigenicity and inducing the humoral immune response [51]. Epitope mapping for ASFV proteins is useful for further understanding host–virus interactions and is important for designing epitope-based vaccines and diagnostic tools. Knowing the antibody–epitope interaction is the basis of the virus prevention measures. Therefore, this study performed epitope mapping analysis using these characterized mAbs and the peptide scanning (Pepscan) method, where multiple overlapping peptides were synthesized followed by Western blot [39,40]. In this study, the epitope mapping analysis showed that the CP312R has two linear antigenic epitopes which are located in the range of ^122^KNEQGEEIYP^131^ (mAb-D7 and mAb-B2), and ^78^DEEVIRMNAE^87^ (mAb-C4, C5, B11, G11 and E4) AA residues. These identified epitopes were detected by their respective mAbs during the IFA analyses in HEK-293T cells. The ssDNA interaction region of CP312R [31] is consistent with previous reports which state that it is conserved among the OB-fold complexes [52]. In the CP312R structural study, the indicated K94, R98, R194, K196, R257, and Y46 AA residues may be important in the ssDNAs’ interaction [31]; K122 and Y122 AA residues of the antigenic epitope of CP312R recognized by the five mAbs, as well as R83 in the second epitope detected by two mAbs, may contribute to the ssDNA interaction.

According to the evidence of several high-resolution structure analyses, the OB-fold proteins interact with nucleotides primarily through aromatic stacking and hydrophobic packing, or the aliphatic portions of more polar groups, such as arginine and lysine [52,53]. In addition to the electrostatic and hydrophobic interactions and disulfide bonds, the protein stability is primarily determined by the AA composition difference factor [54]. Valine (V81), isoleucine (I82 and I129), and Alanine (A86) AA residues in the epitopes of CP312R may also play a vital role in maintaining the structural stability of the ssDNA through the formation of hydrophobicity during its transformation into the dsDNA. These AA residues are a subset of hydrophobic–aliphatic AAs, which are mainly located in the protein core and have carbon-heavy atoms on their side chains. They do not interact physically, and they cluster together to prevent any contact with the solvent. Consequently, hydrophobic interactions are mostly responsible for stabilizing proteins’ shapes [55]. The K122 and R83 AA residues may participate directly in the SSB–ssDNA interactions, whereas the V81, I82, I129, and A86 AA residues contribute to the structural integrity of the ssDNA while it transforms into DNA. Both the antigenic regions are enriched of glutamic acid (E), 30% of the total AA composition of each epitope; any manipulation (substitution/mutation) works consider that the charged amino acids, such as R83, aspartic acid (D78) and glutamic acid (E79, E80, E87, E124, E127, E128), could contribute to the electrostatic interaction, which is a vital force for maintaining conformational stability in the outer part of protein [56,57]. Generally, the epitopes found in the region toward the N-terminal of the identified SSB domain of CP312R protein (61–278 AAs) [31] may be targeted for the development of a vaccine or epitope-based inhibitory antiviral therapeutic options. Because the SSBs are vital in viral DNA replication, repair, and recombination in both eukaryotes and prokaryotes [58,59], and because they interact with other proteins that are able to bind DNA [59,60,61], SSBs regulate viral replication.

The serological diagnosis test is a worthy diagnostic method for pathogen identification through antigen or antibody detection. However, many pathogens, especially similar ones, may have cross-reactions and be unable to clearly discriminate the course of the disease in seroassay tests. As a result, the epitope mapping technology has emerged with the goal of improving the specificity and sensitivity and to eliminate the cross-reaction through designing epitope-based-seroassay tests. Since the epitopes specifically interact with chemical groups that play a role in the antigenicity, their combination has the potential to distinguish different courses of the disease [62]. CP312R is a multistage expressed ASFV protein [50] that has a ssDNA binding site [31]. This study, in addition to verifying the CP312R immunogenicity and mAb production, identified two linear epitopes that are located in the ssDNA binding core region of the CP312R protein. These antigenic determinants are highly conserved across the 21 ASFV strains from seven genotypes (I, II, IV, X, XI, XX, and XXI); genotype VIII, however, has not retained three and two AA residues in the identified ^122^KNEQGEEIYP^131^ and ^78^DEEVIRMNAE^87^ epitopes, respectively. Thus, these valuable findings regarding the CP312R protein may benefit the development of serodiagnosis kits, either in alone or combination with other immunogenic ASFV proteins, and for the development of epitope-based cross-protective vaccines.

Identification of the subcellular location of proteins is useful for studying mechanisms of signal transduction, pathological processes, and cellular metabolism mechanisms [63]. The CP312R localization analysis in virus-infected WSL and PAMs using the characterized mAb-C4 revealed that it was located mainly in the cytoplasm and, to some extent, in the nuclei and on the nuclear membrane of the infected host cells. Its location is likely to have a role in the disruption of communication between the host transcription machinery and the nuclear membrane of the infected host cells. This may help ASFV encode all the factors that are necessary for its mRNA expression and processing during infection [15,64]. Further identification of the peptide signal for the CP312R localization may also give a clue to the development of the control measures for the virus. In conclusion, this study reported the immunogenicity of CP312R, characterized a panel of seven anti-ASFV CP312R mouse mAbs, identified two epitopes of CP312R by using a panel of mAbs, and explored the CP312R subcellular localization during the ASFV infection in its host cell.

## Figures and Tables

**Figure 1 viruses-15-00557-f001:**
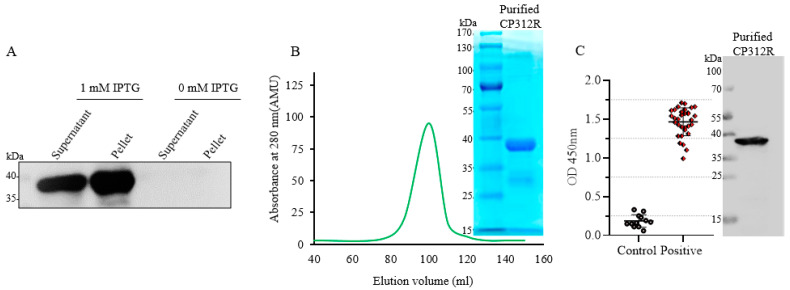
The recombinant CP312R expression in prokaryotic cells and its immunogenicity analysis using sera from pigs infected with the ASFV/HLJ/HRB1/2020 strain. Analysis of the recombinant CP312R protein expression by Western blotting assay using an anti-His-tag antibody (A). The AKAT affinity-based liquid chromatography purification by using a 6x-His Trap HP column process revealed a single peak in the 85–110 mL elution volume. The purified recombinant CP312R protein was subjected to SDS-PAGE for CBB (**B**). The target protein’s immunogenicity was evaluated using the iELISA and Western blot tests (**C**), with ASFV-positive pig sera (*n* = 36, pigs infected with the Pig/HLJ/HRB1/2020 strain) and SPF pig controls (*n* = 12, pigs free from the ASFV).

**Figure 2 viruses-15-00557-f002:**
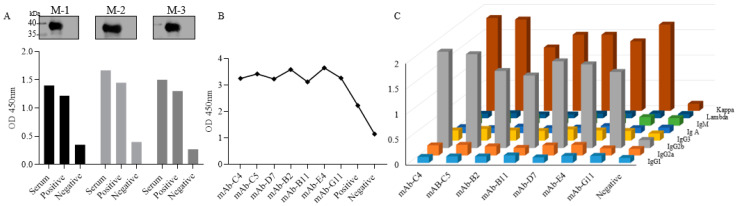
Monoclonal antibody production and isotyping. iELISA and Western blotting analyses were used to screen the mouse for an immune response against the target protein (**A**). A single hybridoma clone’s reactivity analysis was performed by iELISA on the recombinant CP312R protein (**B**). The generated mAbs were isotyped into their immunoglobulin (IG) class (heavy or light chains) by using the Pierce Rapid ELISA Mouse mAb Isotyping Kit (ThermoFisher Scientific) (**C**). M-1 = Mouse number one, M-2 = Mouse number two, M-3 = Mouse number three.

**Figure 3 viruses-15-00557-f003:**
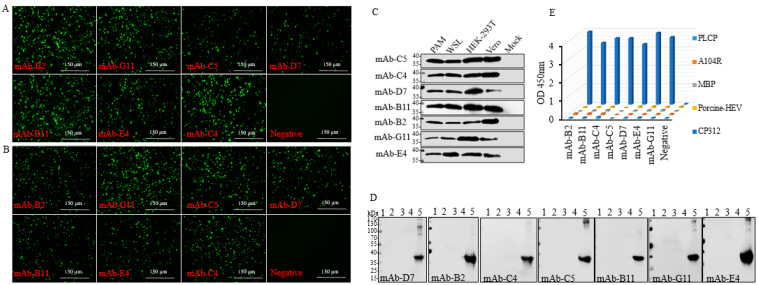
The specificity analyses for mAbs. The IFA analyses were performed in HEK-293T cells transfected with the plasmid of CP312R (1 µg/well, 12-well plate) (**A**) and in PAMs infected with Pig/HLJ/2018 isolate (MOI = 1) (**B**) to observe the CP312R-mAbs’ specific interaction. The generated mAbs interacted with the CP312R protein expressed in various cells infected with the ASFV and lentivirus and transfected with the plasmid CP312R (**C**). The anti-ASFV CP312R mAbs’ specificity was evaluated using a Western blotting assay (**D**) and an iELISA (**E**). The 96-well plates were coated with the purified A104R, PLCP, porcine-HEV, and swine MBP proteins (0.2 µg/mL). Lane 1 = A104R protein, lane 2 = PLCP (Papain-Like Cysteine Protein), lane 3 = Porcine-HEV (Porcine Hepatitis E virus), lane 4 = Swine MBP (Myelin Basic Protein), lane 5 = CP312R protein, WSL=Fetal wild boar lung cell line.

**Figure 4 viruses-15-00557-f004:**
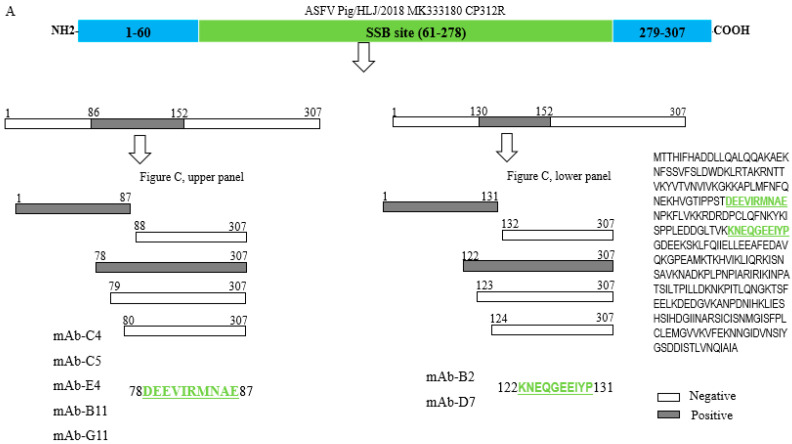
The CP312R antigenic determinants are identified by using the characterized mAbs. The schematic representation of the ASFV CP312R and its truncated fragments in the epitope identification process. The numbers in the boxes represent the positions of the amino acids (AAs) in the CP312R protein as well as the truncated fragment protein (**A**). Identification of the recognizable truncated region using the seven characterized mAbs by Western blot (**B**). Further identification of the epitopes in the recognized truncates using the characterized mAbs by Western blot (**C**). IFA analysis of the identified epitopes by the defined mAbs (**D**).

**Figure 5 viruses-15-00557-f005:**
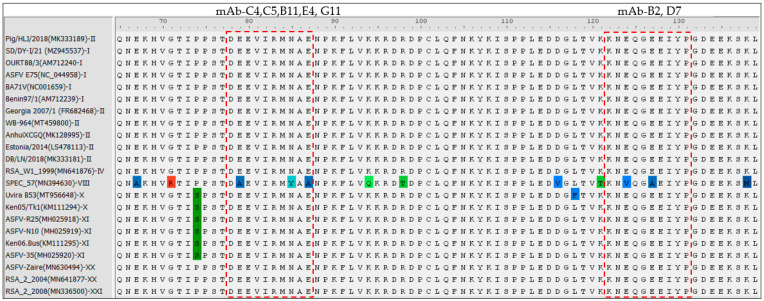
Conservation analysis of newly identified epitopes of the CP312R protein. Sequences of the CP312R proteins from 21 various ASFV strains belonging to eight genotypes were obtained from NCBI database, and aligned using DNAStar software. The red boxes represent the identified epitope sequences. The differently colored amino acids are presented as unconserved amino acids in comparison to the main protein core.

**Figure 6 viruses-15-00557-f006:**
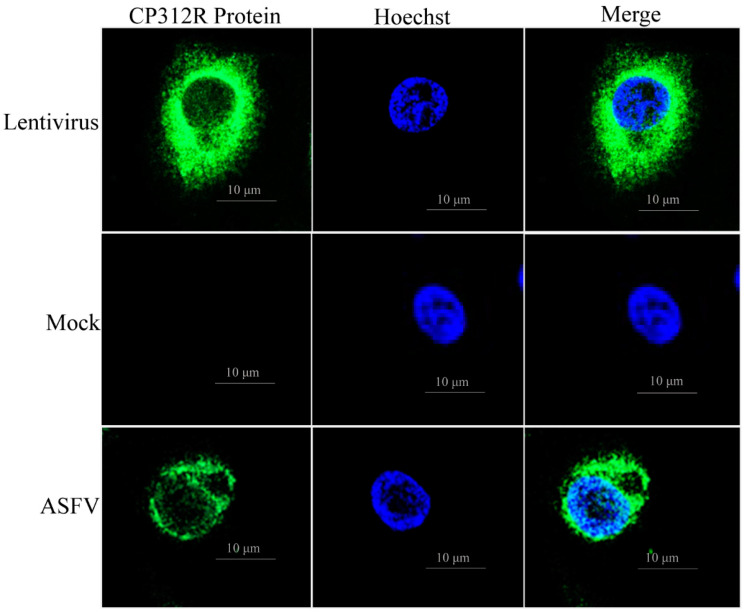
ASFV CP312R subcellular localization analysis in WSL and PAMs infected with the lentivirus-CP312R and ASFV HLJ/HRB1/2020, respectively.

**Table 1 viruses-15-00557-t001:** Primers used for plasmids construction.

Fragments	Primer Sequences (5′-3′)
F1-152	ATCATTTTGGCAAAGAATTGCCACCATGACTACACACATCTTTCACGCAGATG
R1-152	AGGGAAAAAGATCTGCTAGCTCGATTACAAGTCCTCTTCAGAAATGAGCTTTTGCTCGTCTTCAAAGGCTTCTTCTAACAGTTC
R1-141	AGGGAAAAAGATCTGCTAGCTCGATTACAAGTCCTCTTCAGAAATGAGCTTTTGCTCTTGAAACAACTTAGATTTTTCTTCGTCGCCGG
R1-130	AGGGAAAAAGATCTGCTAGCTCGATTACAAGTCCTCTTCAGAAATGAGCTTTTGCTCGTATATTTCTTCACCCTGCTCATTC
R1-119	AGGGAAAAAGATCTGCTAGCTCGATTACAAGTCCTCTTCAGAAATGAGCTTTTGCTCAGTGAGACCATCATCTTCCAATGGCG
R1-108	AGGGAAAAAGATCTGCTAGCTCGATTACAAGTCCTCTTCAGAAATGAGCTTTTGCTCTTTGTATTTGTTGAACTGCAAACAGGGATCC
R1-86	AGGGAAAAAGATCTGCTAGCTCGATTACAAGTCCTCTTCAGAAATGAGCTTTTGCTCAGCATTCATCCGTATAACCTCTTCATCGG
R1-75	AGGGAAAAAGATCTGCTAGCTCGATTACAAGTCCTCTTCAGAAATGAGCTTTTGCTCGGGAGGAATGGTTCCTACATGTTTTTC
R1-64	AGGGAAAAAGATCTGCTAGCTCGATTACAAGTCCTCTTCAGAAATGAGCTTTTGCTCAAAGTTAAACATTAGCGGAGCTTTTTTGCC
F153-307	CATCATTTTGGCAAAGAATTGCCACCATGGTGCAAAAAGGTCCTGAAGCCATGAAAACG
R153-307	AGGGAAAAAGATCTGCTAGCTCGATTACAAGTCCTCTTCAGAAATGAGCTTTTGCTCAGCAATAGCAATCTGATTAACAAGAGTTG
F1-87	AGGGAAAAAGATCTGCTAGCTCGATTACAAGTCCTCTTCAGAAATGAGCTTTTGCTCTTCAGCATTCATCCGTATAACCTCTTCATCGG
R1-87	TCTCATCATTTTGGCAAAGAATTGCCACCATGACTACACACATCTTTCACGCAGATG
R122-307	AGGGAAAAAGATCTGCTAGCTCGATTACAAGTCCTCTTCAGAAATGAGCTTTTGCTCAGCAATAGCAATCTGATTAACAAGAGTTG
F122-307	CATCATTTTGGCAAAGAATTGCCACCATGAAGAATGAGCAGGGTGAAGAAATATACCCCG
F123-307	CATCATTTTGGCAAAGAATTGCCACCATGAATGAGCAGGGTGAAGAAATATACCCCGGCG
F124-307	CATCATTTTGGCAAAGAATTGCCACCATGGAGCAGGGTGAAGAAATATACCCCGGCGACG
F1-131	CATCATTTTGGCAAAGAATTGCCACCATGACTACACACATCTTTCACGCAGATG
R1-131	AGGGAAAAAGATCTGCTAGCTCGATTACAAGTCCTCTTCAGAAATGAGCTTTTGCTCGGGGTATATTTCTTCACCCTGCTCATTC
F132-307	CATCATTTTGGCAAAGAATTGCCACCATGGACGAAGAAAAATCTAAGTTGTTTC
F77-307	CATCATTTTGGCAAAGAATTGCCACCATGACCGATGAAGAGGTTATACGGATGAATGC
F78-307	CATCATTTTGGCAAAGAATTGCCACCATGGATGAAGAGGTTATACGGATGAATGCTG
F79-80	CATCATTTTGGCAAAGAATTGCCACCATGGAAGAGGTTATACGGATGAATGCTGAAAATCC
F80-307	CATCATTTTGGCAAAGAATTGCCACCATGGAGGTTATACGGATGAATGCTGAAAATCC
F88-307	CATCATTTTGGCAAAGAATTGCCACCATGCCAAAGTTTTTGGTGAAAAAACGTGACAGGGATCCC
F141	CATCATTTTGGCAAAGAATTGCCACCATGCAAATTATTGAACTGTTAGAAGAAGCCTTTGAAGACGC
R141	AGGGAAAAAGATCTGCTAGCTCGATTACAAGTCCTCTTCAGAAATGAGCTTTTGCTCAGCAATAGCAATCTGATTAACAAGAGTTG
CP312RHis-F	CGACGATCGATATGACCACCCACATCTTTC
CP312RHis-R	GTAGCTAGCCTCGAGCGCAATCGCAATCTGG
CP312R-F	CATCATTTTGGCAAAGAATTGCCACCATGACTACACACATCTTTCACGCAGATG
CP312R-R	AGGGAAAAAGATCTGCTAGCTCGATTAAGCAATAGCAATCTGATTAACAAGAGTTG

## Data Availability

All information about this research project is included in this article, and for any further reasonable additional data demand, it will be available from the corresponding author.

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
