# Peer review of "Novel Epitopes Mapping of African Swine Fever Virus CP312R Protein Using Monoclonal Antibodies"

_viruses, 2023, doi:10.3390/v15020557_

Round 1

Reviewer 1 Report

In this manuscript Yibrah and his/her colleges identified ASFV monoclonal antibodies, provided detail information of the core region of epitope, and sub location of protein CP312. Considered ASF was the most disease of swine industry, the investigation of basic virology and the basic materials were important for virologist, and provide basic support for the future research of ASF. However, there are some concerns that need to be clarified.

1. The core region of epitope was identified, however the conservation among different ASFVs should be studied, and this will provide evidence for the future usage of the antibody.

2. Some discussion of the future usage(ELISA or others) might be added in the discussion.

Reviewer 2 Report

The MS entitled “Novel Epitopes Mapping of African Swine Fever Virus 2 CP312R Protein Using Monoclonal Antibodies” is an interesting research in the aspect of epitopes mapping of African swine fever virus using Mabs. The MS is well planned and organized and followed the international protocols. The abstract is well written and various molecular studies such as Immunofluorescence Assay (IFA), western blot, Subcellular localization of CP312R and cell transfection studies etc. In results section is very impressive with good quality of figures. The discussion is well discussed with appropriate references. I strongly recommended the MS is for publication   

Author Response

Thank you so much for your positive remarks and strong request that our work be published.

Reviewer 3 Report

In this MS, the authors generated 7 anti-ASFV CP312R mAbs from mice immunized with recombinant CP312R protein, and using the 7 mAbs, they identified two antigenic-determinant regions and the subcellular localization of CP312R. The results provided some valuable information for understanding the antigenic regions of ASFV CP312.

But to the epitope mapping recognized by the mAbs, after the 87-100 and 131-141 AA residues of the CP312R protein were identified by the 7 mAbs, the subsequent experiment was designed incorrectly. For example, to131-141 AA residues, the expressed truncated fragments should be 1-140, 1-139, 1-138, 1-137 …, 132-307, 133-307, 134-307 …, until the truncated expressed protein does not interact with the mAbs. In the results, the 131PGDEEKSKLFQ141 region were recognized by the mAb-B2/D7, and the truncates from 1-132, 1-133 and 1-135 including the control 1-141 fragment were bound specifically with the mAb-B2/D7. According to my expertise, the experimental results of WB may also be wrong, at least, the truncated fragments 1-132 and 1-133 were not bound specifically with the mAb-B2/D7. The same problem occurred with the 87ENPKFLVKKRDRDP100 region. The authors can refer to the article (Zhang, R., et al., Identification of a conserved neutralizing linear B-cell epitope in the VP1 proteins of duck hepatitis A virus type 1 and 3. Veterinary microbiology2015, 180(3-4), 196–204.)

Round 2

Reviewer 3 Report

In this MS, the authors generated 7 anti-ASFV CP312R mAbs from mice immunized with recombinant CP312R protein, and using the 7 mAbs, they identified two antigenic-determinant regions and the subcellular localization of CP312R. The results provided some valuable information for understanding the antigenic regions of ASFV CP312. But the authors should improve the quality of Figure 4A. To be precise, the authors should draw exactly the location of the truncated fragment proteins.
